# Effect of Prior Antibiotic Use on Culture Results in People with Diabetes and Foot Osteomyelitis

**DOI:** 10.3390/antibiotics12040684

**Published:** 2023-03-31

**Authors:** Meryl Cinzía Tila Tamara Gramberg, Jarne Marijn Van Hattem, Jacob Albert Dijkstra, Emma Dros, Max Nieuwdorp, Louise Willy Elizabeth Sabelis, Edgar Josephus Gerardus Peters

**Affiliations:** 1Amsterdam UMC, Vrije Universiteit Amsterdam, Department of Internal Medicine, Section of Infectious Diseases, De Boelelaan 1117, NL-1081HV Amsterdam, The Netherlands; 2Amsterdam Institute for Infection and Immunity and Amsterdam Movement Sciences, P.O. Box 22660, NL-1100DD Amsterdam, The Netherlands; 3Amsterdam UMC, Vrije Universiteit Amsterdam, Department of Rehabilitation Medicine, De Boelelaan 1117, NL-1081HV Amsterdam, The Netherlands; 4Amsterdam UMC Center for Diabetic Foot Complications (ACDC), P.O. Box 22660, NL-1100DD Amsterdam, The Netherlands; 5Amsterdam UMC, Vrije Universiteit Amsterdam, Department of Medical Microbiology and Infection Prevention, Meibergdreef 9, NL-1100DD Amsterdam, The Netherlands; 6Department of Pharmacy and Clinical Pharmacology, Amsterdam University Medical Center, NL-1081HV Amsterdam, The Netherlands; 7Amsterdam UMC, Location University of Amsterdam, Department of Experimental Vascular Medicine, Meibergdreef 9, NL-1100DD Amsterdam, The Netherlands; 8Amsterdam Movement Sciences, P.O. Box 22660, NL-1100DD Amsterdam, The Netherlands

**Keywords:** diabetic foot osteomyelitis, ulcer bed biopsy, bone biopsy, bacterial cultures, antibiotics, antibiotic resistance

## Abstract

Background: Antibiotic use prior to biopsy acquisition in people with diabetes and osteomyelitis of the foot (DFO) might influence bacterial yield in cultures or induce bacterial resistance. Obtaining reliable culture results is pivotal to guide antibiotics for conservative treatment of DFO. Methods: We prospectively analysed cultures of ulcer bed and percutaneous bone biopsies of people with DFO and investigated if antibiotics administered prior to (<2 months up to 7 days) biopsy acquisition led to more negative cultures or increased resistance in virulent bacteria. We calculated relative risks (RR) and 95% confidence intervals (CIs). We stratified analyses according to biopsy type (ulcer bed or bone). Results: We analysed bone and ulcer bed biopsies of 64 people, of whom 29 received prior antibiotics, and found that prior antibiotics did not lead to a higher risk of at least one negative culture (RR 1.3, (CI 0.8–2.0), nor did prior treatment increase the risk of a specific type of negative culture (RR for bone cultures 1.15, (CI 0.75–1.7), RR for ulcer bed cultures 0.92 (CI 0.33–2.6)) or both cultures (RR 1.3 (CI 0.35–4.7), and neither did it increase the risk of antibiotic resistance in the combined bacterial results of ulcer bed and bone cultures (RR 0.64, (CI 0.23–1.8)). Conclusions: Antibiotics administered up to 7 days before obtaining biopsies in people with DFO do not influence culture yield regardless of biopsy type, nor are they associated with more antibiotic resistance.

## 1. Introduction

The cornerstone of conservative treatment of diabetic foot osteomyelitis (DFO) is antibiotic therapy based on susceptibility testing of cultures of an adequately obtained ulcer bed and/or bone biopsy [1,2]. Since people with diabetes suffer frequently from infections, they are often treated with antibiotics [3]. Antibiotic exposure prior to diagnostic biopsy for DFO might influence culture results, but scientific data show conflicting results [4,5,6].

DFO usually starts with a soft tissue or ulcer infection spreading contiguously to the bone [7]. A number of people receive antibiotics for soft tissue infection before the actual bone infection becomes apparent. If clinically safe, guidelines advise interrupting antibiotic therapy prior to obtaining the ulcer bed and bone biopsy for susceptibility testing to guide antibiotic therapy [1,8]. The duration of such an antibiotic-free period differs between 24 h and 2 weeks, depending on the health care setting and doctor preference. If an antibiotic-free period is too short, this prior treatment might lower the overall bacterial yield since the bacterial load is already reduced and antibiotics in the bone sample could possibly hinder growth in vitro [4,5,6]. Antibiotic exposure could induce antibiotic resistance or lead to the selection of bacteria resistant to the previously administered antibiotics [9,10,11]. Results of such cultures might lead to incorrect rejection of the diagnosis of osteomyelitis, or to the misguided prescription of antibiotics aimed at colonising bacteria.

At present, it is unclear if antibiotic use prior to biopsy taking for DFO influences culture results or how long an antibiotic-free period should be before reliable biopsies can be obtained. 

We hypothesised that the administration of antibiotics within two months and up to 7 days prior to obtaining biopsies will lead to more negative cultures and a higher risk of antibiotic resistance in these cultures. To investigate this, we compared the culture results of people with prior antibiotic treatment with those without prior antibiotic treatment with regard to culture yield and antibiotic resistance.

## 2. Results

We included 64 people in this observational study. Table 1 shows demographics and baseline characteristics.

### 2.1. Culture Yield

Of the 64 participants, 36 (56.3%) had at least one negative culture, and in 28 (43.8%), both cultures showed bacterial growth. Of the participants with prior antibiotic treatment, 62.0% had at least one specimen without growth versus 50.0% of participants without prior antibiotic treatment (Table 2).

When calculating relative risks with 95% confidence intervals, prior antibiotic use did not lead to a higher risk of at least one negative culture, nor did prior treatment increase the risk of sterility in a specific biopsy type (bone or ulcer bed) or both types (Table 3). We investigated the effect of an antibiotic-free period of exactly 7 days versus longer than 7 days by adding this as a categorical variable to the logistic regression model. Logistic regression analyses showed that the odds of negative cultures do not differ for participants with an antibiotic-free period of exactly 7 days or longer than 7 days compared to participants without prior treatment. 

When adding the number of antibiotic courses as a categorical variable to the model, the group of people with two prior antibiotic courses was too small, therefore the odds could not be estimated.

### 2.2. Bacterial Resistance

In Table 4 we provide an overview of the total number of identified bacterial species for participants with and without prior antibiotic use and the number of virulent-resistant species.

Of the 64 participants included, 47 (73.3%) had growth of a virulent bacterial species (i.e., *Staphylococcus aureus* and *Staphylococcus lugdunensis*, beta-haemolytic streptococci, *Enterobacterales,* or *Pseudomonas* spp.) in either one of the biopsies. In 14 (29.8%) of these 47 participants, we identified at least one resistant species.

When calculating relative risks with 95% confidence intervals, prior antibiotics did not increase the risk of culturing resistant bacteria (Table 5). The calculated relative risk would even suggest a lower risk in people with prior antibiotic exposure.

We investigated the effect of an antibiotic-free period of exactly 7 days versus longer than 7 days by adding this categorical variable to the logistic regression model. Logistic regression analyses showed that the relation between antibiotic exposure and resistance is different for participants with an antibiotic-free period longer than 7 days compared to participants with an antibiotic-free period of exactly 7 days. For participants with an antibiotic free period longer than 7 days, we found odds lower than 1, implying that these participants had lower odds of resistant cultures compared to participants without prior treatment. We found odds higher than 1 in participants with an antibiotic free period of exactly 7 days, implying that the odds of resistant cultures are higher than in participants without prior treatment. 

Logistic regression analyses showed that for participants with prior treatment with two courses, the odds of resistant cultures are higher than for participants with only one course of antibiotics compared to participants without prior treatment (Table 5).

### 2.3. Bacterial Selection

Considering the bacterial species identified in cultures of participants with and without prior treatment, we found several differences. We identified other gram-positive bacteria and corynebacteria more frequently in participants with prior antibiotic treatment. In participants without prior treatment, we identified *Staphylococcus aureus*, *coagulase-negative staphylococci* (CNS)/*Enterococcus* spp., and anaerobes more frequently (Table 4).

We identified 65 bacterial species in 29 participants with prior treatment (average of 2.4 unique bacteria per person) versus 89 bacterial species in 35 participants without prior treatment (average of 2.5 unique bacteria per person). We did not identify a lower number of bacteria in participants without prior treatment due to bacterial selection.

## 3. Discussion

We examined the impact of previous antibiotic use on the bacterial yield and resistance in ulcer bed and bone biopsy cultures from persons with diabetes and foot osteomyelitis. We found that administering antibiotics up to 7 days before obtaining biopsies did not affect culture yield or increase the risk of antibiotic resistance, regardless of the type of culture. In the literature, there is controversy regarding the influence of prior antibiotics on culture yield. Young et al. retrospectively investigated the effect of pre-operative antibiotics on operatively-obtained soft tissue or bone specimens of people with diabetic foot infection [6]. They found that antibiotic exposure was associated with less frequent growth of streptococci and anaerobes and more frequent culture-negative results, but no impact on the growth of *S aureus*, *Enterococcus* species, and Enterobacteriaceae. That study probably obtained different results compared to our study because intravenous antibiotics were administered in the 7 days prior to biopsy, in contrast with our participants who received oral antibiotics up to 7 days prior to biopsy and did not receive antibiotics during the 7 days directly preceding the biopsy. When comparing bacterial species, Young et al. identified more *Streptococcus* spp. and anaerobes in people with fewer hours of total antibiotic exposure, and we identified more *Streptococcus* (non-beta-haemolytic and beta-haemolytic taken together) and anaerobes in participants without prior antibiotics. Different to Young et al., we did identify more *S. aureus* in people without prior antibiotics, whereas Young et al. did not. Two abstracts reported contradictory results regarding the influence of antibiotic exposure prior to bone biopsy on culture yield in people with osteomyelitis [12,13]. One abstract reported that a longer duration of antibiotic exposure was negatively correlated to culture positivity in people with osteomyelitis, of whom, 48% had lower extremity osteomyelitis [12]. The other specifically investigated people with DFO and reported that culture yield was similar for people with and without prior treatment regardless of the duration of antibiotic exposure [13]. Both abstracts investigated the duration of antibiotic exposure as a categorical variable in days of antibiotic exposure, but neither specified the timeframe in which antibiotics were administered before biopsy acquisition [12,13].

Besides some literature regarding lower extremity osteomyelitis, there are also some studies in vertebral osteomyelitis showing contradictory results [5,14,15,16]. People in these studies received antibiotics (including intravenous antibiotics) at different moments before biopsy acquisition (ranging from hours to days before biopsy), and results varied from no impact to significant impact on culture yield [5,14,15,16].

A strength of our study is that we investigated the influence of prior treatment on ulcer bed and bone cultures separately and combined. By doing this, we investigated several clinical scenarios (e.g., sterile bone and/or ulcer bed cultures, or only sterile bone, but positive ulcer bed cultures). Bone biopsy is not common practice in all hospitals for the diagnosis and identification of pathogens in DFO. Ulcer bed biopsies are also obtained in case of soft tissue infection without osteomyelitis. Investigating the yield of bone and ulcer bed biopsies separately makes the results applicable for both soft tissue and bone infection. Additionally, by comparing both sampling methods, we eliminated the inequality of the pre-test likelihood of positive ulcer bed and bone cultures. Ulcer bed cultures are open to surroundings, which increases the pre-test likelihood that cultures of the ulcer bed become positive compared with bone. 

We did not find an increased risk for negative cultures in participants with prior antibiotic use with relatively small 95% confidence intervals, suggesting a low uncertainty of this finding. When we stratified participants according to the antibiotic-free period (exactly 7 days or longer than 7 days), groups became small with concomitant wider 95% confidence intervals, indicating a larger uncertainty of the obtained results, and one of the stratified groups even became too small to reliably estimate results with logistic regression analyses. 

Based on our results, an interval of at least 7 days might be safe, but larger randomised controlled trials should confirm these findings, and these further studies could investigate if an even shorter interval might also be feasible.

In the present study, we did not find an association between prior antibiotics and the occurrence of resistant bacteria in either bone or ulcer bed cultures, although we investigated resistance according to a predefined clear definition based on guidelines and antimicrobial stewardships for the treatment of DFO [17,18,19,20].

Our results showed a different relation between antibiotic exposure and resistance for participants with an antibiotic-free period longer than 7 days compared to participants with an antibiotic-free period of exactly 7 days in comparison to participants without prior antibiotic treatment. Based on the pathophysiological mechanisms of resistance induction and the literature, we do not believe that prior antibiotics can have a protective effect on resistance induction [9,11,21,22,23,24]. However, there are data that suggest that peripheral artery disease is associated with a higher likelihood of antimicrobial resistance [12].

We tried to investigate the influence of the number of antibiotic courses on the occurrence of resistance because we assumed that this variable could be of influence [6,13,25]. Therefore, we also wanted to investigate the influence of the absolute dose of antibiotics administered to participants on the occurrence of resistance. Unfortunately, the absolute amount of administered antibiotics differed too much within our population; therefore, we could not overcome the problem of small groups and multiple testing. 

Because of our clear and unambiguous definition of resistance, we could not classify all culture results, i.e., cultures only consisting of uncommon gram-positive or -negative bacteria, anaerobes, or bacteria, which are usually intrinsically resistant (e.g., *Corynebacterium* spp.). These unclassified results were, therefore, not taken into account in the analyses, and we had to accept some loss of power for this part of the study. This is reflected by the larger 95% confidence intervals in the logistic regression analyses.

## 4. Materials and Methods

### 4.1. Objective and Hypothesis

The aim of this study was to evaluate the effect of prior antibiotics on culture results of ulcer and bone biopsy in people with DFO.

We evaluated if antibiotic use within 2 months and up to 7 days prior to biopsy lead to:○A lower yield of conventional cultures (any growth versus no growth);○A higher risk of antibiotic resistance in cultures, with a focus on virulent bacteria. 

We hypothesised that the administration of antibiotics shortly before obtaining biopsies will lead to more negative cultures, and that prior antibiotics lead to a higher risk of antibiotic resistance in virulent bacteria in these cultures.

### 4.2. Design

This is a prospective observational study, using data from cultures taken during the international multicentre randomised controlled BeBoP trial [2]. Through the informed consent procedure of the BeBoP study, participants gave consent for further studies, including the present study. 

The BeBoP trial included participants (18 ≥ years) with diabetes mellitus and DFO in several sites in The Netherlands and Australia, from which, participant data from the Amsterdam University Medical Centres, locations Meibergdreef and De Boelelaan, were used for the present study.

### 4.3. Study Procedures

We included participants with suspected DFO, based on a positive probe-to-bone test, abnormalities on plain X-ray suggestive of osteomyelitis [26], erythrocyte sedimentation rate (ESR) ≥ 70 mm/h (without another explanation for the elevated ESR) [1,27], signs of osteomyelitis on MRI and/or FDG-PET/CT-scan [26], positive microbiological or molecular culture results, or histology of a recent percutaneous bone biopsy performed before inclusion.

All participants underwent an aseptically obtained percutaneous bone biopsy, adjacent to the ulcer through intact sterilised skin and a biopsy of the ulcer bed at inclusion. Specimens were examined using conventional culturing techniques for bacteria, according to standard operating procedures of the Amsterdam University Medical Centres laboratory. A Gram stain was performed and bacteria were inoculated on Columbia agar + 5% sheep blood (COS) and chocolate agar (PVX) at 35–37 °C under aerobic conditions with carbon dioxide (CO_2_). Besides these aerobic cultures, all specimens were also inoculated under anaerobic conditions, i.e., inoculation of bacteria on COS at 35–37 °C for 4 days. Specimens were inoculated in brain heart infusion broth (BHI) for 7 days to increase sensitivity. If these cultures showed bacterial growth, they were further inoculated on PVX and COS at 35–37 °C with CO_2_ under both aerobic and anaerobic conditions.

### 4.4. Prior Antibiotic Treatment

We collected participant data regarding antibiotic consumption from 2 months up to 7 days prior to the acquisition of ulcer bed and bone biopsies from: 1. The Electronic Health Record (EHR) of Amsterdam University Medical Centres; 2. The National Pharmacy Database (*Landelijk Schakelpunt*, LSP). In our database, we scored whether a participant received antibiotics within the given timeframe of 2 months up to 7 days before biopsies, as well as the type of antibiotic agents, the number of different antibiotic agents, and the number of antibiotic courses. We excluded a course of nitrofurantoin and a single dose of oral fosfomycin, prescribed for uncomplicated urinary tract infection. These antibiotics do not reach therapeutic levels in the skin or bone. We also scored if a participant received prior antibiotics up to exactly 7 days before the acquisition of cultures or if the antibiotic-free period was longer. We created this variable to investigate if there is a difference between the risk of antibiotics administered shortly before biopsy or at a longer period of time prior.

### 4.5. Culture Results

The microbiologist scored whether a culture was negative, and which type of specimen was sterile (ulcer bed and/or bone). A negative culture was defined as the complete absence of growth, and a positive one included any bacterial growth, including the growth of low-virulence bacteria such as coagulase-negative staphylococci, *Corynebacterium* spp., and enterococci.

The microbiologist determined if the combined results of ulcer bed and bone cultures of one participant were either susceptible (i.e., all bacteria in both cultures were susceptible to key antibiotics) or resistant (i.e., at least one bacterium in one of the cultures was resistant to key antibiotics). We investigated the occurrence of resistance of common virulent (pathogenic) bacteria in DFO: *S. aureus* and *S. lugdunensis*, beta-haemolytic streptococci, *Enterobacterales*, and *Pseudomonas* spp. to one or more key antibiotic(s) for the treatment of DFO.

Resistance to key antibiotics was defined as:Clindamycin-resistance in *S. aureus*, *S. lugdunensis,* and beta-haemolytic streptococci;Ciprofloxacin-resistance in *Enterobacterales* and *Pseudomonas* spp.;Trimethoprim/sulfamethoxazole-resistance in *Enterobacterales;*Multiple drug resistance (MDR), defined as acquired resistance to at least one agent in three or more antimicrobial classes [26].

According to the new definition of S/I/R by EUCAST, isolates that are tested in the ‘I’ category are considered susceptible (i.e., not resistant) [18].

We explored the occurrence of intrinsically resistant bacteria in cultures after the administration of prior antibiotics (pre-exposed vs. not pre-exposed participants): *Corynebacterium* spp., coagulase-negative staphylococci, and *Enterococcus* spp.

### 4.6. Analyses

To investigate if prior antibiotics increased the risk of sterile cultures or increased the risk of culturing resistant bacteria in the cultures, we calculated relative risks with 95% confidence intervals. We calculated relative risks for the outcomes: 1. at least one sterile culture, 2. bone culture negative, 3. ulcer bed culture negative, 4. both cultures are negative, 5. resistant bacteria/a resistant bacterium in the cultures. We combined the obtained relative risks and rates of uncertainty with an estimation of clinical relevance. Additionally, we performed logistic regression analyses to investigate the influence of a shorter (7 days before acquisition of specimens) compared to a longer (>7 days but <2 months before acquisition of specimens) antibiotic-free interval. To investigate this, we created a categorical variable with three categories (no prior treatment, treatment < 2 months > 7 days, antibiotic-free period of exactly 7 days) and added this to the model. We also investigated the influence of the number of antibiotic courses prior to biopsies. Therefore, we created a categorical variable: 0 courses, 1 course, or 2 courses, and added this to the logistic regression model.

## Figures and Tables

**Table 1 antibiotics-12-00684-t001:** Demographics and baseline characteristics.

	N	Minimum	Maximum	Median	Percentage
Demographics					
Total people included	64				
Age (years)	64	39	95	67	
Duration of diabetes (years)	59	4	55	20	
Gender (male)	48				75.0
≥1 missing pedal pulsation	33				51.6
Baseline characteristics					
Antibiotics prior to biopsy ^1^	29				45.3 ^8^
Antibiotics up to exactly 7 days before biopsy ^2^	14				21.8 ^8^
Number of antibiotic courses ^3^					
0	24				53.1 ^8^
1	21				25.0 ^8^
2	8				10.9 ^8^
Number of different antibiotic agents ^4^					
0	24				53.1 ^8^
1	19				29.7 ^8^
2	8				12.5 ^8^
3	2				3.1 ^8^
Antibiotic agents ^5^					
Flucloxacillin	13				44.8 ^9^
Amoxicillin (clavulanic acid) ^6^	4				13.8 ^9^
Clindamycin	5				17.2 ^9^
Ciprofloxacin	5				17.2 ^9^
Other ^7^	2				6.9 ^9^

^1^ The number of participants who received antibiotics up to two months before biopsies were obtained. ^2^ The number of participants receiving antibiotics up to exactly 7 days before biopsies were obtained, i.e., 15 participants had a longer antibiotic-free interval. ^3^ The number of antibiotic courses a participant received in the 2 months before biopsy acquisition. ^4^ The number of different antibiotic agents administered to a participant during the 2 months before biopsy acquisition. ^5^ Type of antibiotic agent administered to a participant during the 2 months before biopsy acquisition. ^6^ One participant was treated with amoxicillin and 3 participants with amoxicillin/clavulanic acid. ^7^ One participant was treated with vancomycin and 1 participant was treated with trimethoprim/sulfamethoxazole. ^8^ Percentages of a total of 64 participants. ^9^ Percentage of participants with prior antibiotic treatment, N = 29.

**Table 2 antibiotics-12-00684-t002:** Culture results.

Outcomes	Prior Antibiotic Use		Total Number of Participants
	Yes (N = 29)	No (N = 35)	
At least one negative culture ^1^	18 (50%)	18 (50%)	36
Negative bone cultures ^2^	5 (42%)	7 (58%)	12
Negative ulcer bed cultures ^3^	17 (47%)	19 (53%)	36
Both negative ^4^	4 (44%)	5 (56%)	9

^1^ The number of participants with at least one negative culture. ^2^ The number of participants with negative bone cultures. ^3^ The number of participants with negative ulcer bed cultures. ^4^ The number of participants with negative bone and ulcer bed cultures.

**Table 3 antibiotics-12-00684-t003:** Relative risks and odds for negative cultures in people with prior antibiotic use.

Outcomes	Relative Risk	95% Confidence Interval	
At least one negative culture or both	1.3	0.84	2.0
Negative bone cultures	1.1	0.75	1.7
Negative ulcer bed cultures	0.92	0.33	2.6
Both negative	1.3	0.35	4.7
	Odds ^1^	95% Confidence interval	
At least one negative culture or both			
AFP > 7 days	2.1	0.60	7.5
AFP = 7 days	1.9	0.53	6.8
Negative bone cultures			
AFP > 7 days	1.4	0.42	4.8
AFP = 7 days	1.7	0.47	6.1
Negative ulcer bed cultures			
AFP > 7 days	0.3	0.03	2.6
AFP = 7 days	1.6	0.39	6.7
Both negative			
AFP > 7 days	NE ^2^		
AFP = 7 days	2.4	0.54	10.7

AFP = Antibiotic Free Period; NE = Not Estimated; ^1^ The odds of negative culture(s) in participants with prior antibiotic use, outcomes of logistic regression analyses. Analyses are stratified for the AFP, i.e., an AFP of more than 7 days (AFP > 7 days) or an AFP of 7 days (AFP = 7 days). Participants without prior antibiotics are the reference category, therefore all results are in comparison to participants without prior antibiotics (i.e., participants with an AFP of 7 days have odds of 1.9 compared to participants without prior antibiotics of having at least one negative culture or both). ^2^ The group of participants with negative cultures and an AFP > 7 days was too small (N = 7), therefore odds could not be estimated.

**Table 4 antibiotics-12-00684-t004:** Culture results according to prior antibiotic use.

Prior Antibiotic Use	Yes (N = 29)		No (N = 35)	
Bacteria Cultured	Number ^1^ (Resistant)	Relative Percentage ^2^	Number ^1^ (Resistant)	Relative Percentages ^2^
Non-beta-haemolytic *Streptococcus*	3	10.3	8	22.9
Beta-haemolytic *Streptococcus*	6 (1)	20.7	7 (1)	20.0
*Staphylococcus aureus*	12 (1)	41.4	20 (4)	57.1
MRSA	0	0	1 (1)	2.9
*Staphylococcus lugdunensis*	3 (1)	10.3	1 (0)	2.9
CNS/*Enterococcus* spp.	6	20.7	14	40.0
*Corynebacterium* spp.	6	20.7	4	11.4
*Enterobacterales*	15 (3)	51.2	20 (4)	57.1
*Pseudomonas* spp.	2 (0)	6.9	4 (0)	11.4
Gram positive other ^3^	7	24.1	1	2.9
Gram negative other ^4^	3	10.3	3	8.6
Anaerobes ^5^	2	6.9	6	17.1

MRSA = Methicillin Resistant *S. aureus*. CNS=Coagulase negative *Staphylococcus* ^1^ The number of times bacterial species were cultured in participants with(out) prior antibiotic treatment. Within brackets, the number of times bacterial species were resistant to at least one or more of the prespecified common antibiotics for DFO. ^2^ The relative percentage bacterial species cultured per group, e.g., non-beta-haemolytic streptococci were cultured 3 times in participants with prior antibiotic treatment. The total number of participants with prior antibiotic treatment was 29. Therefore, we calculated the percentage of 3 of 29. Non-beta-haemolytic streptococci were cultured 8 times in participants without prior antibiotic treatment. The total number of participants without prior antibiotic treatment is 35. Therefore, we calculated the percentage of 8 per 35 people. ^3^ Other gram-positive bacteria including *Dermabacter hominis*, *Aerococcus urinae*, *Parvimonas micra*, *Kocuria* spp., *Brevibacterium casei*, and unspecified gram-positive bacteria. ^4^ Other gram-negative bacteria including *Haemophilus parainfluenzae*, *Enterobacter cloacae* complex, *Aeromonas* spp. and *Pasteurella dagmatis*. ^5^ Anaerobes including: *Finegoldia magna* and *Bacteroides fragilis, Actinomyces odontolitica*, *Prevotella* spp., and unspecified anaerobes.

**Table 5 antibiotics-12-00684-t005:** Relative risk and odds for resistant cultures in participants with prior antibiotic use.

Outcomes	Relative Risk for Resistance	95% Confidence Interval	
	0.64	0.24	1.7
	Odds ^1^	95% Confidence interval	
AFP > 7 days	0.79	0.17	3.7
AFP = 7 days	1.2	0.18	7.8
1 course ^2^	1.3	0.27	5.9
2 courses ^3^	1.5	0.18	12.8

AFP = Antibiotic-free period. ^1^ The odds of resistant culture(s) in participants with prior antibiotic use, outcomes of logistic regression analyses. Analyses are stratified for the AFP, i.e., an AFP of more than 7 days (AFP > 7 days) or an AFP of 7 days (AFP = 7 days). Participants without prior antibiotics (=0 courses) are the reference category; therefore, all results are in comparison to participants without prior antibiotics (i.e., participants with an AFP of 7 days have odds of 1.2 compared to participants without prior antibiotics to have resistant cultures, and participants with 2 prior courses have odds of 1.5 compared to participants without prior antibiotics to have cultures). ^2^ after one course of antibiotics. ^3^ after two courses of antibiotics.

## Data Availability

Data regarding this study are provided in the Appendix A sections. After closure of the BeBoP trial, data will become available in a data repository which will then be created.

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
