# Peer review of "Effect of Prior Antibiotic Use on Culture Results in People with Diabetes and Foot Osteomyelitis"

_antibiotics, 2023, doi:10.3390/antibiotics12040684_

Round 1

Reviewer 1 Report

Dear authors, thank you very much for the opportunity to review your well described research.

The study describes a prospective cohort of 64 persons with diabetic foot osteomyelitis treated with antibiotic divided by prior antibiotic regimen < 7 days and > 7 days. The authors diagnosed DFO via probe to bone test and radiological examination in addition to blood test. As primary outcome measure they evaluated if prior ATB increased the risk of resistant bacterias.

The results of the current research suggest that time of prior atb regimen in OM patients do not influence the results of the culture nor even resistances.

The introduction is well described and includes a brief state of the art for the studied topic. 

Research design and methods are well designed, OM diagnosis is based on the IWGDF guideliness. I just have a minor query with the sample size calculation. Was this calculated? If no, I think authors should calculate the power calculation for the sample size in order to improve the external and internal validity of the research. The N of the research seems too low.

Results are given in mean and standard deviation and median IQR depending on the normal and non-normal distribution. Was the distribution of the sample evaluated in statistics? If not please revise.

Discussion: the results of the paper was compared with previous literature. Strengths are well defined.

Despite this, if authors finally did not calculate sample size it should be stated as a limitation. Additionally, the results of the current research should be confirmed in a RCT basis.

Congrats on your paper once again.

Reviewer 2 Report

First of all I would like to congratulate the authors on this thoughtful and well written paper. The study question arises multiple times per week in my daily practice as a diabetic foot surgeon and I am glad I had a chance to review it.

While I do think that the manuscript is sound, there are minor details that should be corrected:

Table 1: Please add "years" to age and duration of diabetes.

Table 1: Please provide information on PAD and its severity in your study population. It may influence antibiotic susceptibility (Małecki R, Klimas K, Kujawa A. Different Patterns of Bacterial Species and Antibiotic Susceptibility in Diabetic Foot Syndrome with and without Coexistent Ischemia. J Diabetes Res. 2021 Apr 27;2021:9947233. doi: 10.1155/2021/9947233. PMID: 34007849; PMCID: PMC8100368.). Add that to your discussion, also.

Table 2: Please add percentages after the absolute numbers like you did in Table 4.

Lines 212-214: Syntax problem, please rephrase.
